# A New Efficient and Provably Secure Certificateless Signature Scheme Without Bilinear Pairings for the Internet of Things

**DOI:** 10.3390/s25175224

**Published:** 2025-08-22

**Authors:** Zhanzhen Wei, Xiaoting Liu, Hong Zhao, Zhaobin Li, Bowen Liu

**Affiliations:** Department of Electronic and Communication Engineering, Beijing Electronic Science and Technology Institute, Beijing 100070, China

**Keywords:** certificateless signature, Internet of Things, forgery attack

## Abstract

**Highlights:**

**What are the main findings?**
A new attack method, the *Common Factor Substitution Attack*, is introduced, showing that a broad class of existing pairing-free certificateless signature (PF-CLS) schemes share a common vulnerability and cannot resist Type I adversary forgeries—even after certain prior improvements.A new PF-CLS scheme is proposed that securely binds both parts of a user’s public key to the system public key via a hash function, eliminating the Type II attack vulnerability in Ma et al.’s scheme, and is proven secure against both Type I and Type II adversaries under the Random Oracle Model while offering higher computational efficiency than comparable schemes.

**What is the implication of the main finding?**
The results underscore that many PF-CLS schemes in current use for resource-constrained environments like IoT remain insecure against forged signature attacks unless both Type I and Type II threats are systematically addressed in their design.The proposed scheme’s combined security and efficiency make it a strong candidate for real-world IoT deployments, improving trust in data integrity and identity authentication without imposing heavy computational or communication costs.

**Abstract:**

Pairing-free certificateless signature (PF-CLS) schemes are ideal authentication solutions for resource-constrained environments like the Internet of Things (IoT) due to their low computational, storage, and communication resource requirements. However, it has come to light that many PF-CLS schemes are vulnerable to forged signature attacks. In this paper, we use a novel attack method to prove that a class of PF-CLS schemes with the same security vulnerabilities cannot resist Type I adversary attacks, and we find that, even if some schemes are improved to invalidate existing attack methods, they still cannot defend against the new attack method proposed in this paper. Subsequently, we introduce an enhanced scheme proven to be resilient against both types of adversary attacks under the random oracle model (ROM). Performance analysis shows that, compared with several existing PF-CLS schemes, our scheme offers higher computational efficiency.

## 1. Introduction

The IoT is used broadly and, because of its openness, transmits most data over the public Internet, exposing it to eavesdropping, tampering, forgery, and other attacks. Consequently, data security is paramount for IoT. Digital signature technology, ensuring the authentication, unforgeability, and non-repudiation of messages, is ideal for IoT environments.

In a traditional public key cryptosystem, an entity’s public key’s authenticity and validity require a digital certificate from a trusted third-party, the Certificate Authority (CA). Managing certificates in IoT environments is challenging because of the substantial overhead for storing, distributing, verifying, and revoking them. In Identity-Based Cryptography (IBC), a user’s public key is generated based on identifiable public information, like a phone number, email address, or IP address. A trusted third-party Private Key Generator (PKG) creates the user’s Private Key, eliminating the need for CAs and digital certificates and addressing the certificate management issue. However, since the PKG fully generates the user’s private key, this raises concerns over potential signature forgeries, undermining the unforgeability of the system. Certificateless Public Key Cryptography (CL-PKC) solves the key escrow issue in IBC by having a Key Generation Center (KGC) create a partial private key for the user. The user then combines this with their own data to generate a full private key.

Consequently, CLS amalgamates the strengths of traditional public key cryptosystem and IBC, addressing their respective drawbacks and proving more effective for ensuring data integrity and identity authentication in IoT’s distributed, flexible, and resource-limited environments.

### 1.1. Related Work

In 2003, AI-Riyami and Paterson proposed the first certificateless signature scheme [1]. This scheme, which does not require digital certificates, allows the user and the KGC to jointly generate both public and private keys. The KGC only holds a portion of the user’s private keys, effectively solving the key escrow problem inherent in IBC. In 2004, Yum et al. proposed a general construction for CLS schemes [2]. This led to the development of numerous CLS schemes [3,4]. However, these schemes all relied on bilinear pairings, which are known for their high computational complexity and low efficiency. In 2011, He et al. introduced the first CLS scheme that does not use bilinear pairings [5]. However, Tian and Tsai later pointed out that He’s scheme was vulnerable to Type II adversary attacks [6,7]. In 2013, Gong et al. proposed a pairing-free CLS scheme [8], but Yeh et al. pointed out that it could not resist Type I adversary attacks and proposed an improved version [9]. In 2015, Wang et al. introduced a more efficient variant of Yeh et al.’s scheme [10]. In 2017, Yeh et al. discovered vulnerabilities in Wang’s improved scheme regarding forged signatures and subsequently developed a new scheme [11]. However, Jia et al. later revealed that Yeh’s new scheme was still vulnerable to Type II adversary attacks involving public key replacement [12]. In 2018, Karati et al. independently proposed a new scheme that does not use bilinear pairings and claimed that it can resist both Type I and Type II adversary attacks under the ROM [13]. Later, Pakniat et al. successfully demonstrated that Karati et al.’s scheme can be attacked by Type I adversaries, and they improved the scheme [14]. Shim et al. found that Karati et al.’s scheme allows for forging arbitrary message signatures without obtaining any secret information, and that Pakniat et al.’s scheme still cannot resist Type I adversary attacks [15]. In the same year, Du et al. pointed out that Jia et al.’s improved scheme cannot resist Type I adversary attacks and subsequently developed an enhanced version [16]. Thumbur et al. independently proposed a new pairing-free certificateless signature scheme [17], but it was later pointed out by Xu et al. [18] that it cannot resist Type I adversary public key replacement attacks. By 2022, Xiang et al. successfully demonstrated a specific Type II adversary attack method targeting Jia et al.’s scheme and improved it based on identified vulnerabilities [19]. In 2023, Ma et al. pointed out that both Du et al.’s and Xiang et al.’s schemes are susceptible to Type I adversary attacks, and proposed new attack methods and improvements [20]. However, Feng et al. [21] found that Ma et al.’s scheme still could not resist Type II adversary attacks the following year.

### 1.2. Our Contributions

This paper’s primary contributions are as follows:

1. We analyze a class of PF-CLS schemes with the same security vulnerabilities and use a new attack method to show that all schemes in this class cannot resist Type I adversary attacks.

2. We introduce a new pairing-free certificateless signature scheme, which tightly binds the two parts of the user’s public key ($X_{ID}$ and $R_{ID}$) to the system’s public key ($P_{pub}$) via a hash function, avoiding the vulnerability of Type II adversary attack due to independent generation of public keys in the scheme of Ma et al. [20]. Moreover, our scheme proves its security against both Type I and Type II adversary attacks under the Random Oracle Model (ROM).

3. Performance evaluations demonstrate that our scheme has higher computational efficiency compared to other people’s schemes, making it more suitable for IoT environments.

## 2. Preliminaries

### 2.1. Complexity Assumption

Consider *q* as a large prime number, with G representing an additive group of order *q* over the elliptic curve $E/F_{q}$, where *P* is a generator of G. The Elliptic Curve Discrete Logarithm Problem (ECDLP) means that, for any instance $Q\;=\;aP,\;a\in Z_{q}^{*},Q\in G$, if P,Q are known, finding the solution yields *a*.

The difficult hypothesis for the Elliptic Curve Discrete Logarithm Problem is that there is no probabilistic polynomial algorithm *A* that can solve ECDLP with a non-negligible probability in polynomial time.

### 2.2. Security Model

Certificateless signature schemes face two types of attacks from adversaries. An adversary of Type I ($A_{I}$) has the capability to substitute the user’s public key without requiring access to the system master key. On the other hand, an adversary of Type II ($A_{II}$) holds possession of the system master key but does not possess authority to alter or replace the user’s public key. The security model is represented as a gaming scenario involving both challengers $C_{1}\left(C_{2}\right)$ and their respective adaptive adversaries $A_{I}\left(A_{II}\right)$

**Game 1**: This game is played between an adaptive adversary $A_{I}$ and the challenger $C_{1}$. $C_{1}$ maintains a list $L_{1}$ that stores information about users. The game consists of the following phases:

**Initialization**: Challenger $C_{1}$ enters a security parameter *k*, generates system parameters params, and sends them to $A_{I}$, while $C_{1}$ saves the system master key *s* itself.

**Query phase**: In this phase, the following queries can be submitted adaptively by $A_{I}$. That is, the adversary will issue a new query based on the received response.

1. CreateUser query: Enter user identity (ID); $C_{1}$ queries list $L_{1}$. If the user has been created, $C_{1}$ returns its corresponding user public key ($PK_{ID}$); otherwise, $C_{1}$ performs three algorithms, ExtractPartialPrivateKey, ExtractSecretValue, and SetPublicKey, to create a user and returns the $PK_{ID}$. It then adds the ID, partial private key ($d_{ID}$), secret value ($x_{ID}$), and $PK_{ID}$ to list $L_{1}$ for maintenance.

2. ReplacePublicKey query: Enter $\left(ID,PK_{ID}^{\prime }\right)$; $C_{1}$ queries table $L_{1}$. If the user exists, then $C_{1}$ updates the corresponding public key $PK_{ID}$ in the record table $L_{1}$ as $PK_{ID}^{\prime }$, and records the secret value $x_{ID}$ as ⊥; otherwise, $C_{1}$ executes the CreateUser query first, and then performs the update.

3. ExtractSecretValue query: Enter ID; $C_{1}$ first queries table $L_{1}$. If the user does not exist, then $C_{1}$ executes CreateUser query and returns the corresponding secret value $x_{ID}$; if the user exists and the public key $PK_{ID}$ is not replaced, $C_{1}$ outputs the corresponding secret value $x_{ID}$. Otherwise, output ⊥.

4. ExtractPartialPrivateKey query: Enter ID; $C_{1}$ queries table $L_{1}$. If the user is present, $C_{1}$ returns the corresponding partial private key $d_{ID}$. Otherwise, $C_{1}$ first executes the CreateUser query and then returns $d_{ID}$.

5. SuperSign query: Input (ID,m); $C_{1}$ outputs a signature τ. The signature can be verified by the corresponding message plaintext *m*, system parameter params, and user public key $PK_{ID}$. The $PK_{ID}$ is the last updated public key corresponding to the ID in list $L_{1}$, that is, the public key may not be original.

**Forgery**: After successfully executing all queries, adversary $A_{I}$ generates a counterfeit signature $\left(ID^{*},m^{*},\tau^{*}\right)$. The victory condition for $A_{I}$ is as follows: 1. $A_{I}$ has never queried the super signature for $\left(ID^{*},m^{*}\right)$; 2. $A_{I}$ has never performed an ExtractPartialPrivateKey query for $ID^{*}$; 3. $\tau^{*}$ for $\left(ID^{*},m^{*},PK^{*}\right)$ is a verifiable valid signature, and the public key $PK^{*}$ may not be original.

**Game 2**: This game is played between an adaptive adversary $A_{II}$ and the challenger $C_{2}$.$C_{2}$ maintains a list $L_{2}$ that stores information about users. The game consists of the following phases:

**Initialization**: Challenger $C_{2}$ enters a security parameter *k*, generates system parameters params, and sends them to $A_{II}$, while $C_{2}$ sends the system master key *s* to $A_{II}$.

**Query phase**: In this phase, the following queries can be submitted adaptively by $A_{II}$.

1. CreateUser query: Enter user identity (ID); $C_{2}$ queries list $L_{2}$. If the user has been created, $C_{2}$ returns its corresponding user public key ($PK_{ID}$); otherwise, $C_{2}$ performs three algorithms, ExtractPartialPrivateKey, ExtractSecretValue, and SetPublicKey, to create a user and returns the $PK_{ID}$, and then adds the ID, partial private key ($d_{ID}$), secret value ($x_{ID}$), and $PK_{ID}$ to list $L_{2}$ for maintenance.

2. ExtractSecretValue query: Enter ID; $C_{2}$ first queries table $L_{2}$. If the user does not exist, then $C_{2}$ executes CreateUser query and returns the corresponding secret value $x_{ID}$; if the user exists and the public key $PK_{ID}$ is not replaced, $C_{2}$ outputs the corresponding secret value $x_{ID}$. Otherwise, output ⊥.

3. ExtractPartialPrivateKey query: Enter ID; $C_{2}$ queries table $L_{2}$. If the user is present, $C_{2}$ returns the corresponding partial private key $d_{ID}$. Otherwise, $C_{2}$ first executes the CreateUser query and then returns $d_{ID}$.

4. SuperSign query: Input (ID,m); $C_{2}$ outputs a signature τ. The signature can be verified by the corresponding message plaintext *m*, system parameter params, and user public key $PK_{ID}$. The $PK_{ID}$ is the last updated public key corresponding to the ID in list $L_{2}$, that is, the public key may not be original.

**Forgery**: After successfully executing all queries, adversary $A_{II}$ generates a counterfeit signature $\left(ID^{*},m^{*},\tau^{*}\right)$. The victory condition for $A_{II}$ is as follows: 1. $A_{II}$ has never queried the super signature for $\left(ID^{*},m^{*}\right)$; 2. $A_{II}$ never performs an ExtractSecretValue query on $ID^{*}$; 3. $\tau^{*}$ is a verifiable valid signature for $\left(ID^{*},m^{*},PK^{*}\right)$, and the public key $PK^{*}$ remains intact.

For a CLS scheme, if the super Type I or Type II adversary cannot win Game 1 or Game 2 by a significant advantage in polynomial time, we say that the scheme is unfalsifiable against adaptive chosen plaintext attacks and identity attacks.

## 3. Review of Four PF-CLS Schemes

In this section, we review the four PF-CLS schemes [13,14,16,19]. The four PF-CLS schemes contain six basic process algorithms: Setup, Partial Private Key Extract, Set Secret Value, Set Public/Private Key, Signature, and Verification.

### 3.1. Xiang et al.’s Scheme

Xiang et al.’s scheme [19] contains the following six algorithms:

1. Setup: The KGC generates an elliptic curve additive group G of order *q* and selects P∈G as the generator of group G. The KGC randomly chooses $s\overset{\$}{\leftarrow }Z_{q}^{*}$ as the system master key, computes $P_{pub}\;=\;sP$, and selects three Hash functions: $H_{i(i\;=\;1,2,3)}:{\left\{0,1\right\}}^{*}\to Z_{q}^{*}$. Subsequently, the KGC publishes the system parameters $params\;=\;\{q,G,P,P_{pub},H_{1},H_{2},H_{3}\}$ and keeps the master key *s* secret.

2. Partial Private Key Extract: The KGC takes the system parameters params and identity ID as input. Then, it randomly selects $r_{ID}\overset{\$}{\leftarrow }Z_{q}^{*}$, computes $R_{ID}\;=\;r_{ID}P$, $h_{1}\;=\;H_{1}\left(ID,R_{ID},P_{pub}\right)$, and $d_{ID}\;=\;r_{ID}\;+\;h_{1}s\;\mathrm{mod}\;q$. KGC sends the partial private key $D_{ID}\;=\;\left(R_{ID},d_{ID}\right)$ to the user over a secure channel.

3. Set Secret Value: The user randomly picks $x_{ID}\overset{\$}{\leftarrow }Z_{q}^{*}$ as his/her secret value and calculates $X_{ID}\;=\;x_{ID}P$.

4. Set Public/Private Key: The user sets $PK_{ID}\;=\;\left(R_{ID},X_{ID}\right)$ as the public key and $sk_{ID}\;=\;\left(d_{ID},x_{ID}\right)$ as the private key.

5. Signature: The signer takes the message *m*, params, ID, $PK_{ID}$, and $sk_{ID}\;=\;\left(d_{ID},x_{ID}\right)$ as input, then generates a signature σ on message *m* as follows:

(i) Randomly select a value $t\overset{\$}{\leftarrow }Z_{q}^{*}$, and compute T = tP;

(ii) Compute $h_{2}\;=\;H_{2}\left(ID,T,PK_{ID}\right)$, $h_{3}\;=\;H_{3}\left(ID,m,T,PK_{ID},P_{pub}\right)$, and $\tau \;=\;x_{ID}^{-1}\left(h_{2}t\;+\;h_{3}d_{ID}\right)\;\mathrm{mod}\;q$;

(iii) Output σ = (T,τ) as a signature.

6. Verification: The verifier takes the message–signature tuple (m,σ = (T,τ)), params, ID, and $PK_{ID}\;=\;\left(R_{ID},X_{ID}\right)$ as input, then verifies the signature σ as follows:

(i) Compute $h_{1}\;=\;H_{1}\left(ID,R_{ID},P_{pub}\right)$, $h_{2}\;=\;H_{2}\left(ID,T,PK_{ID}\right)$, $h_{3}\;=\;H_{3}\left(ID,m,T,PK_{ID},P_{pub}\right)$;

(ii) Output “Accept” if $\tau X_{ID}\;=\;h_{2}T\;+\;h_{3}\left(R_{ID}\;+\;h_{1}P_{pub}\right)$ holds, and “Reject” otherwise.

### 3.2. Du et al.’s Scheme

The Setup, Partial Private Key Extract, Set Secret Value, and Set Public/Private Key algorithms in Du et al.’s scheme [16] are basically the same as those in Xiang et al.’s scheme [19], with the only difference being the value of $h_{1}\;=\;H_{1}\left(ID,R_{ID},P,P_{pub}\right)$.

Signature: Given the message *m*, params, ID, $PK_{ID}$, and $sk_{ID}\;=\;\left(d_{ID},x_{ID}\right)$ as input, the signature generation process is as follows:

1. Randomly select a value $t\overset{\$}{\leftarrow }Z_{q}^{*}$, and compute T = tP.

2. Compute $h_{2}\;=\;H_{2}\left(m,T,ID,PK_{ID},P_{pub}\right)$, $h_{3}\;=\;H_{3}\left(m,T,ID,PK_{ID},h_{2}\right)$, and $\tau \;=\;t^{-1}\left(h_{2}x_{ID}\;+\;h_{3}d_{ID}\right)\;\mathrm{mod}\;q$.

3. Output σ = (T,τ) as the signature.

Verification: Given *m*, σ, params, ID, and $PK_{ID}$ as input, the verification process is as follows:

1. Compute $h_{1}\;=\;H_{1}\left(ID,R_{ID},P,P_{pub}\right)$, $h_{2}\;=\;H_{2}\left(m,T,ID,PK_{ID},P_{pub}\right)$, $h_{3}\;=\;H_{3}\left(m,T,ID,PK_{ID},h_{2}\right)$.

2. Output "Accept" if $\tau T\;=\;h_{2}X_{ID}\;+\;h_{3}\left(R_{ID}\;+\;h_{1}P_{pub}\right)$ holds, and "Reject" otherwise.

### 3.3. Karati et al.’s Scheme

Karati et al.’s scheme [13] contains the following six algorithms:

1. Setup: The KGC generates an elliptic curve additive group G of order *q* and selects P∈G as the generator of group G. The KGC randomly chooses $s\overset{\$}{\leftarrow }Z_{q}^{*}$ as the master key, computes $P_{pub}\;=\;\left\{P_{pub1},P_{pub2}\right\}\;=\;\left\{sP,s^{-1}P\right\}$, and selects two Hash functions: $H_{i(i\;=\;1,2)}:{\left\{0,1\right\}}^{*}\to Z_{q}^{*}$. Subsequently, the KGC publishes the system parameters $\mathrm{params}\;=\;(q,G,P,P_{pub},H_{1},H_{2})$ and keeps the master key *s* secretly.

2. Partial Private Key Extract: The KGC takes params and identity ID as input, randomly selects $r_{ID}\overset{\$}{\leftarrow }Z_{q}^{*}$, computes $R_{ID}\;=\;r_{ID}P$, $h_{1}\;=\;H_{1}\left(ID,R_{ID},P\right)$, $d_{ID}\;=\;\left(s^{-1}\;+\;r_{ID}^{-1}sh_{1}\right)\;\mathrm{mod}\;q$, and $Q_{ID}\;=\;r_{ID}s^{-1}P$. KGC sends the partial private key $D_{ID}\;=\;\left(R_{ID},d_{ID},Q_{ID}\right)$ to the user over a secure channel.

3. Set Secret Value: The user randomly selects $x_{ID}\overset{\$}{\leftarrow }Z_{q}^{*}$ as his/her secret value.

4. Set Public/Private Key: The user computes $X_{ID}\;=\;x_{ID}R_{ID}$, then sets $PK_{ID}\;=\;\left(R_{ID},Q_{ID},X_{ID}\right)$ as the public key and $sk_{ID}\;=\;\left(d_{ID},x_{ID}\right)$ as the private key.

5. Signature: The signer inputs *m*, params, ID, $PK_{ID}$, and $sk_{ID}$, then generates signature σ for *m* as follows:

(i) Randomly select a value $t\overset{\$}{\leftarrow }Z_{q}^{*}$, and compute $T\;=\;tR_{ID}$;

(ii) Compute $h_{2}\;=\;H_{2}\left(m⊕ID,T,Q_{ID}\right)$, and $\tau \;=\;x_{ID}{\left(t\;+\;h_{2}x_{ID}\;+\;d_{ID}\right)}^{-1}\;\mathrm{mod}\;q$;

(iii) Output σ = (T,τ) as a signature.

6. Verification: The verifier inputs (m,σ = (T,τ)), params, ID, and $PK_{ID}\;=\;\left(R_{ID},Q_{ID},X_{ID}\right)$, then verifies the signature σ as follows:

(i) Compute $h_{1}\;=\;H_{1}\left(ID,R_{ID},P\right)$, $h_{2}\;=\;H_{2}\left(m⊕ID,T,Q_{ID}\right)$, and $R^{\prime }\;=\;T\;+\;h_{2}X_{ID}\;+\;Q_{ID}\;+\;h_{1}P_{pub1}$;

(ii) Output “Accept” if $X_{ID}\;=\;\tau R^{\prime }$ holds, and “Reject” otherwise.

### 3.4. Pakniat and Vanda’s Scheme

The Setup, Partial Private Key Extract, Set Secret Value, and Set Public/Private Key algorithms in Pakniat and Vanda’s scheme [14] are basically the same as those in Karati et al.’s scheme [13], except that Pakniat and Vanda’s scheme additionally defines an extra Hash function $H_{3}:{\left\{0,1\right\}}^{*}\to Z_{q}^{*}$.

Signature: Given *m*, params, ID, $PK_{ID}$, and $sk_{ID}$ as input, the signature generation process is as follows:

1. Randomly select a value $t\overset{\$}{\leftarrow }Z_{q}^{*}$ and compute $T\;=\;tR_{ID}$.

2. Compute $h_{2}\;=\;H_{2}\left(H_{3}\left(m\right)⊕H_{3}\left(ID\right),T,Q_{ID},0\right)$, $h_{3}\;=\;H_{2}\left(H_{3}\left(m\right)⊕H_{3}\left(ID\right),T,Q_{ID},1\right)$, and $\tau \;=\;x_{ID}{\left(h_{2}t\;+\;h_{3}x_{ID}\;+\;d_{ID}\right)}^{-1}\;\mathrm{mod}\;q$.

3. Output σ = (T,τ) as a signature.

Verification: Given *m*, σ, params, ID, and $PK_{ID}$ as input, the verification process is as follows:

1. Compute $h_{1}\;=\;H_{1}\left(ID,R_{ID},P\right)$, $h_{2}\;=\;H_{2}\left(H_{3}\left(m\right)⊕H_{3}\left(ID\right),T,Q_{ID},0\right)$, $h_{3}\;=\;H_{2}\left(H_{3}\left(m\right)⊕H_{3}\left(ID\right),T,Q_{ID},1\right)$, and $R^{\prime }\;=\;h_{2}T\;+\;h_{3}X_{ID}\;+\;Q_{ID}\;+\;h_{1}P_{pub1}$.

2. Output “Accept” if $X_{ID}\;=\;\tau R^{\prime }$ holds, and “Reject” otherwise.

## 4. Novel Attacks on Four Schemes

In this section, we use a novel attack method on the four schemes presented in Section 3 and prove that all four schemes cannot resist Type I adversary attacks.

### 4.1. Cryptanalysis of Xiang et al.’s and Du et al.’s Schemes

Ma et al. [20] have pointed out that Xiang et al.’s scheme [19] cannot resist Type I adversary attacks, and the specific attack method will not be elaborated here. In this section, we use a new attack method different from Ma et al. to prove that Xiang et al.’s scheme cannot resist Type I adversary attacks. Suppose that there exists a Type I adversary $A_{I}$ attempting to forge the signature of a user with identity ID and public key $PK_{ID}\;=\;\left(R_{ID},X_{ID}\right)$. The process of forging the signature is as follows:

1. $A_{I}$ selects two random values $x_{ID}^{*}\overset{\$}{\leftarrow }Z_{q}^{*}$, $r_{ID}^{*}\overset{\$}{\leftarrow }Z_{q}^{*}$, computes $X_{ID}^{*}\;=\;x_{ID}^{*}P_{pub}$, $R_{ID}^{*}\;=\;r_{ID}^{*}P_{pub}$, and replaces the original public key $PK_{ID}\;=\;\left(R_{ID},X_{ID}\right)$ with $PK_{ID}^{*}\;=\;\left(R_{ID}^{*},X_{ID}^{*}\right)$.

2. Assuming that the message is $m^{*}$, $A_{I}$ randomly selects $t^{*}\overset{\$}{\leftarrow }Z_{q}^{*}$, and computes $T^{*}\;=\;t^{*}P_{pub}$, $h_{1}^{*}\;=\;H_{1}\left(ID,R_{ID}^{*},P_{pub}\right)$, $h_{2}^{*}\;=\;H_{2}\left(ID,T^{*},PK_{ID}^{*}\right)$, $h_{3}^{*}\;=\;H_{3}\left(ID,m^{*},T^{*},PK_{ID}^{*},P_{pub}\right)$, and $\tau^{*}\;=\;{\left(x_{ID}^{*}\right)}^{-1}\left(h_{2}^{*}t^{*}\;+\;h_{3}^{*}\left(r_{ID}^{*}\;+\;h_{1}^{*}\right)\right)$. Then, $A_{I}$ outputs the forged signature $\sigma^{*}\;=\;\left(T^{*},\tau^{*}\right)$.

3. Upon receiving the message–signature tuple $\left(m^{*},\sigma^{*}\;=\;\left(T^{*},\tau^{*}\right)\right)$, params, and $PK_{ID}^{*}\;=\;\left(R_{ID}^{*},X_{ID}^{*}\right)$, the verifier first computes $h_{1}^{*}\;=\;H_{1}\left(ID,R_{ID}^{*},P_{pub}\right)$, $h_{2}^{*}\;=\;H_{2}\left(ID,T^{*},PK_{ID}^{*}\right)$, $h_{3}^{*}\;=\;H_{3}\left(ID,m^{*},T^{*},PK_{ID}^{*},P_{pub}\right)$, and then verifies the equation $\tau^{*}X_{ID}^{*}\;=\;h_{2}^{*}T^{*}\;+\;h_{3}^{*}\left(R_{ID}^{*}\;+\;h_{1}^{*}P_{pub}\right)$.

We can observe that the forged signature can easily pass the verification equation:$$\begin{array}{cc} \tau^{*}X_{ID}^{*} & \;=\;{\left(x_{ID}^{*}\right)}^{-1}\left(h_{2}^{*}t^{*}\;+\;h_{3}^{*}\left(r_{ID}^{*}\;+\;h_{1}^{*}\right)\right)x_{ID}^{*}P_{pub}\\ & \;=\;h_{2}^{*}t^{*}P_{pub}\;+\;h_{3}^{*}\left(r_{ID}^{*}\;+\;h_{1}^{*}\right)P_{pub}\\ & \;=\;h_{2}^{*}T^{*}\;+\;h_{3}^{*}\left(R_{ID}^{*}\;+\;h_{1}^{*}P_{pub}\right) \end{array}$$

Through the above attack method, the attacker can forge a valid signature for any user on any message. Therefore, Xiang et al.’s scheme cannot resist Type I adversary attacks. Moreover, the attack on Du et al.’s scheme [16] is similar to the attack on Xiang et al.’s scheme, and will not be elaborated here.

### 4.2. Cryptanalysis of Pakniat and Vanda’s and Karati et al.’s Schemes

Shim [15] pointed out that Pakniat and Vanda’s scheme [14] cannot resist Type I adversary attacks, and the specific attack method will not be elaborated here. In this section, we first improve Pakniat and Vanda’s scheme to make Shim’s attack method ineffective. Then, we use the same attack method as in Section 4.1 to demonstrate that the improved Pakniat and Vanda’s scheme still cannot resist Type I adversary attacks.

The method for improving Pakniat and Vanda’s scheme is relatively simple, requiring only a change in the calculation of $h_{1}$ from $h_{1}\;=\;H_{1}\left(ID,R_{ID},P\right)$ to $h_{1}\;=\;H_{1}\left(ID,R_{ID},P,Q_{ID},P_{pub1}\right)$. The improved scheme will not be described in detail here. After this improvement, if we still want to forge $Q_{ID}^{*}\;=\;-h_{1}P_{pub1}$, we need to provide the value of $h_{1}$ to obtain $Q_{ID}^{*}$, and, in order to obtain the value of $h_{1}$,we need to first provide the value of $Q_{ID}^{*}$. This is contradictory, so we cannot forge and Shim’s attack cannot succeed.

Next, we prove that the improved Pakniat and Vanda’s scheme still cannot resist Type I adversary attacks. Suppose that a Type I adversary $A_{1}$ attempts to forge the signature of a user with identity ID and public key $PK_{ID}\;=\;\left(R_{ID},Q_{ID},X_{ID}\right)$. The detailed process is as follows:

1. $A_{1}$ selects three random values $x_{ID}^{*}\overset{\$}{\leftarrow }Z_{q}^{*}$, $r_{ID}^{*}\overset{\$}{\leftarrow }Z_{q}^{*}$, $q_{ID}^{*}\overset{\$}{\leftarrow }Z_{q}^{*}$, computes $X_{ID}^{*}\;=\;x_{ID}^{*}P_{pub1}$, $R_{ID}^{*}\;=\;r_{ID}^{*}P_{pub1}$, $Q_{ID}^{*}\;=\;q_{ID}^{*}P_{pub1}$, and replaces the original public key $PK_{ID}\;=\;\left(R_{ID},Q_{ID},X_{ID}\right)$ with $PK_{ID}^{*}\;=\;\left(R_{ID}^{*},Q_{ID}^{*},X_{ID}^{*}\right)$.

2. Supposing that the message is $m^{*}$, $A_{1}$ randomly selects $t^{*}\overset{\$}{\leftarrow }Z_{q}^{*}$, and computes $T^{*}\;=\;t^{*}P_{pub1}$, $h_{1}^{*}\;=\;H_{1}\left(ID,R_{ID}^{*},P,Q_{ID}^{*},P_{pub1}\right)$, $h_{2}^{*}\;=\;H_{2}\left(H_{3}\left(m^{*}\right)⊕H_{3}\left(ID\right),T^{*},Q_{ID}^{*},0\right)$, $h_{3}^{*}\;=\;H_{2}\left(H_{3}\left(m^{*}\right)⊕H_{3}\left(ID\right),T^{*},Q_{ID}^{*},1\right)$, $\tau^{*}\;=\;x_{ID}^{*}{\left(h_{2}^{*}t^{*}\;+\;h_{3}^{*}x_{ID}^{*}\;+\;q_{ID}^{*}\;+\;h_{1}^{*}\right)}^{-1}\;\mathrm{mod}\;q$. Then, $A_{1}$ outputs the forged signature $\sigma^{*}\;=\;\left(T^{*},\tau^{*}\right)$.

3. After receiving the message–signature pair $\left(m^{*},\sigma^{*}\;=\;\left(T^{*},\tau^{*}\right)\right)$, params, and $PK_{ID}^{*}\;=\;\left(R_{ID}^{*},Q_{ID}^{*},X_{ID}^{*}\right)$, the verifier first computes $h_{1}^{*}\;=\;H_{1}\left(ID,R_{ID}^{*},P,Q_{ID}^{*},P_{pub1}\right)$, $h_{2}^{*}\;=\;H_{2}\left(H_{3}\left(m^{*}\right)⊕H_{3}\left(ID\right),T^{*},Q_{ID}^{*},0\right)$, $h_{3}^{*}\;=\;H_{2}\left(H_{3}\left(m^{*}\right)⊕H_{3}\left(ID\right),T^{*},Q_{ID}^{*},1\right)$, and then verifies the equation $\tau^{*}\left(h_{2}^{*}T^{*}\;+\;h_{3}^{*}X_{ID}^{*}\;+\;Q_{ID}^{*}\;+\;h_{1}^{*}P_{pub1}\right)\;=\;X_{ID}^{*}$.

We can observe that the forged signature can easily pass the verification equation:$$\begin{array}{cc} \tau^{*}( & h_{2}^{*}T^{*}\;+\;h_{3}^{*}X_{ID}^{*}\;+\;Q_{ID}^{*}\;+\;h_{1}^{*}P_{pub1})\\ \;=\; & x_{ID}^{*}{\left(h_{2}^{*}t^{*}\;+\;h_{3}^{*}x_{ID}^{*}\;+\;q_{ID}^{*}\;+\;h_{1}^{*}\right)}^{-1}\left(h_{2}^{*}t^{*}\;+\;h_{3}^{*}x_{ID}^{*}\;+\;q_{ID}^{*}\;+\;h_{1}^{*}\right)P_{pub1}\\ \;=\; & x_{ID}^{*}P_{pub1}\\ \;=\; & X_{ID}^{*} \end{array}$$

Through the above attack method, the attacker can forge a valid signature for any user on any message. Therefore, Pakniat and Vanda’s scheme cannot resist Type I adversary attacks. Moreover, we can use the same attack method to prove that Karati et al.’s scheme [13] cannot resist Type I adversary attacks, and the detailed process can be referred to above, which will not be repeated here.

**Definition 1.** 
*In our attack, $X_{ID}^{*}$, $R_{ID}^{*}$, $T^{*}$ are generated by random values $x_{ID}^{*}$, $r_{ID}^{*}$, $t^{*}$, which do not have a relationship with each other, and therefore the verifier cannot detect the forgery of the public key and signature. Compared with the attack method of Ma et al., our attack method is practical. We call this novel attack the Common Factor Substitution Attack.*


## 5. The Proposed CLS Scheme

In this section, we introduce a novel CLS scheme that offers enhanced security and efficiency (Figure 1).

1. Setup: KGC takes the security parameter *k* as input and selects the elliptic curve addition group G, where *q* and *P* are the order and generator of the group G, respectively. KGC chooses two secure hash function $H_{1}:{\left\{0,1\right\}}^{*}\;\times \;G^{3}\to Z_{q}^{*}$, $H_{2}:{{\left\{0,1\right\}}^{*}}^{2}\;\times \;G^{3}\to Z_{q}^{*}$. Then, KGC randomly selects the system master key $s\overset{\$}{\leftarrow }Z_{q}^{*}$, and calculates the system public key $P_{pub}\;=\;sP$. $params\;=\;(G,q,P,P_{pub},H_{1},H_{2})$ will be outputted as the public system parameters.

2. SetSecretValue: User identified by ID randomly chooses the secret value $x_{ID}\overset{\$}{\leftarrow }Z_{q}^{*}$, then computes $X_{ID}\;=\;x_{ID}P$ and transmits it to KGC.

3. PartialPrivateKeyExtract: After receiving $\left(ID,X_{ID}\right)$ submitted by the user, KGC randomly selects $r_{ID}\overset{\$}{\leftarrow }Z_{q}^{*}$ and calculates $R_{ID}\;=\;r_{ID}P$ as part of the public key, then calculates $h_{1}\;=\;H_{1}\left(ID,X_{ID},R_{ID},P_{pub}\right)$. Finally, KGC calculates $d_{ID}\;=\;r_{ID}\;+\;h_{1}s$ as the partial private key and transmits $\left(d_{ID},R_{ID}\right)$ to the user via a secure communication channel.

4. SetPrivateKey: The user generates the complete private key $SK_{ID}\;=\;\left(x_{ID},d_{ID}\right)$.

5. SetPublicKey: The user generates the complete public key $PK_{ID}\;=\;\left(X_{ID},R_{ID}\right)$.

6. Sign: The signer signs the message *m*. The signer randomly picks $t\overset{\$}{\leftarrow }Z_{q}^{*}$ and calculates T = tP. And then the signer calculates $h_{2}\;=\;H_{2}\left(m,ID,X_{ID},R_{ID},T\right),$$v\;=\;t\;+\;h_{2}\left(x_{ID}\;+\;d_{ID}\right)\;mod\;q$ to obtain signature τ = (T,v)

7. Verify: The verifier receives signature τ and message *m* for verification. First, the verifier calculates $h_{1}\;=\;H_{1}\left(ID,X_{ID},R_{ID},P_{pub}\right),$$h_{2}\;=\;H_{2}\left(m,ID,X_{ID},R_{ID},T\right)$, and then determines whether the verification equation $vP\;=\;T\;+\;h_{2}\left(X_{ID}\;+\;R_{ID}\;+\;h_{1}P_{pub}\right)$ is valid. It accepts if true, and rejects the signature otherwise.

## 6. Security Proof

**Forking Lemma:** In cryptography, the forking lemma is a crucial tool for proving the security of digital signature schemes. Its core idea is as follows: if there exists an adversary capable of successfully forging a valid signature triple δ = (m,σ,h) (where *m* denotes the message, σ denotes the signature value, *h* denotes the hash value generated by the random oracle *H*, and it contains a publicly accessible value *z*), then, by replaying the interaction process between the adversary and the random oracle, the adversary can be forced to generate a different hash output $\left(h^{*}\right)$ for the same input, thereby obtaining another valid signature $\delta^{*}\;=\;\left(m,\sigma^{*},h^{*}\right)$.

Specifically, by fixing the adversary’s random tape and resetting the random oracle, the adversary can be compelled to generate two distinct yet valid signatures during repeated executions. These two signatures satisfy $h\;\neq \;h^{*}$, and their existence can be used to construct an algorithm to solve the underlying cryptographic hard problem (such as the discrete logarithm problem) with a non-negligible probability. The advantage probability of the forking lemma is $\left(1\;-\;\frac{1}{e}\right)\cdot \frac{\varepsilon }{q_{h}}$, where ε denotes the original forgery probability of the adversary, $q_{h}$ denotes the number of oracle queries, and *e* denotes the natural constant.

**Theorem 1.** 
*Suppose that the probability that a super Type I adversary $A_{I}$, being able to produce a valid forged signature in polynomial time, is ε, the maximum number of times that an adversary $A_{I}$ can query CreateUser query is $q_{H_{1}}$, and the maximum number of times that an adversary $A_{I}$ can query the ExtractPartialPrivateKey query is $q_{eppk}$. The polynomial time algorithm $C_{1}$ has the advantage of $\varepsilon_{1}\;\geq \;{\left(1\;-\;\frac{1}{q_{H_{1}}}\right)}^{q_{eppk}}\frac{1}{q_{H_{1}}}\varepsilon$ to solve difficult ECDLP problems.*


**Proof.** Suppose that the super Type I adversary can win Game 1 by a non-negligible advantage in polynomial time. For a given ECDLP hard problem instance (P,aP), we can construct the algorithm $C_{1}$ to call $A_{I}$ as a subroutine and use the power of $A_{I}$ to solve it. $H_{1}$ and $H_{2}$ are two random oracle models. $C_{1}$ maintains two lists, $L_{1}$ and $L_{2}$, for recording $A_{I}$’s query of $H_{1}$ and $H_{2}$, respectively. $L_{1}$, $L_{2}$ are initially empty. The interaction between $A_{I}$ and $C_{1}$ is as follows:**Initialization phase**: $C_{1}$ inputs security parameter *k* to generate system parameters and sets $P_{pub}\;=\;aP$ (*a* is unknown to $C_{1}$). Then, $C_{1}$ selects the challenge user $ID^{*}$ and sends the system parameters $\left(G,q,P,P_{pub},H_{1},H_{2}\right)$ to the adversary $A_{I}$.**Query phase**:1. CreateUser query: $A_{I}$ enters $ID_{i}$; $C_{1}$ first looks up list $L_{1}$. If $\left(ID_{i},\;x_{i},d_{i},X_{i},R_{i},h_{1i}\right)\in L_{1}$ already exists, $C_{1}$ returns $PK_{i}\;=\;\left(Q_{i},R_{i}\right)$; otherwise, carry out the following: (1) If the $ID_{i}\;\neq \;ID^{*}$, $C_{1}$ randomly selects $x_{i},d_{i},h_{1i}\in Z_{q}^{*}$, calculates $X_{i}\;=\;x_{i}P,$$R_{i}\;=\;d_{i}P\;-\;h_{1i}P_{pub}$, sets $h_{1i}\;=\;H_{1}\left(ID_{i},X_{i},R_{i},P_{pub}\right)$, and returns $PK_{i}\;=\;\left(X_{i},R_{i}\right)$. $C_{1}$ adds $\left(ID_{i},x_{i},d_{i},X_{i},R_{i},h_{1i}\right)$ to table $L_{1}$. (2) If the $ID_{i}\;=\;ID^{*}$, $C_{1}$ randomly selects $x_{i},r_{i},h_{1i}\in Z_{q}^{*}$, calculates $X_{i}\;=\;x_{i}P,$$R_{i}\;=\;r_{i}P$, sets $h_{1i}\;=\;H_{1}\left(ID_{i},X_{i},R_{i},P_{pub}\right)$, and returns $PK_{i}\;=\;\left(X_{i},R_{i}\right)$. $C_{1}$ adds $\left(ID_{i},x_{i},⊥,X_{i},R_{i},h_{1i}\right)$ to list $L_{1}$.2. $H_{1}\;query$: $A_{I}$ inputs $\left(ID_{i},X_{i},R_{i},P_{pub}\right)$. $C_{1}$ first queries list $L_{1}$. If $\left(ID_{i},x_{i},d_{i},X_{i},R_{i},h_{1i}\right)\in L_{1}$ already exists, then $C_{1}$ returns $h_{1i}$ to $A_{I}$; otherwise, $C_{1}$ performs the CreateUser query and then returns $h_{1i}$.3. $H_{2}\;query$: $A_{I}$ inputs $\left(m_{i},ID_{i},X_{i},R_{i},T_{i}\right)$; $C_{1}$ first inquires about $L_{2}$. If $\left(m_{i},ID_{i},X_{i},R_{i},T_{i},h_{2i}\right)\in L_{2}$ already exists, then $C_{1}$ returns $h_{2i}$; otherwise, $C_{1}$ randomly selects $h_{2i}\in Z_{q}^{*},\left(*,*,*,*,*,h_{2i}\right)\notin L_{2}$ and returns $h_{2i}$ to $A_{I}$. Then, $C_{1}$ updates $\left(m_{i},ID_{i},X_{i},R_{i},T_{i},h_{2i}\right)$ to list $L_{2}$.4. ExtractPartialPrivateKey query: $A_{I}$ inputs $ID_{i}$. If $ID_{i}\;=\;ID^{*}$, $C_{1}$ aborts the program and outputs ⊥; otherwise, $C_{1}$ checks list $L_{1}$ and returns $d_{i}$. If $ID_{i}$ does not exist in table $L_{1}$, the CreateUser query will be executed first and then $d_{i}$ will be returned.5. ExtractSecretValue query: $A_{I}$ enters $ID_{i}$. $C_{1}$ queries list $L_{1}$ and returns $x_{i}$. If $ID_{i}$ does not exist in list $L_{1}$, $C_{1}$ executes CreateUser query and then returns $x_{i}$. In addition, if the public key is not the original one, then $C_{1}$ outputs ⊥.6. ReplacePublicKey query: $A_{I}$ inputs $\left(ID_{i},PK_{i}^{\prime }\right)$. $C_{1}$ looks up list $L_{1}$ and updates in the table corresponding to the content as $\left(ID_{i},x_{i},d_{i},X_{i},R_{i},h_{1i}\right)\to \left(ID_{i},⊥,d_{i},X_{i}^{\prime },R_{i}^{\prime },h_{1i}\right)$.7. SuperSign query: $A_{I}$ inputs $\left(ID_{i},m_{i}\right)$. $C_{1}$ checks whether $\left(ID_{i},X_{i},R_{i},P_{pub},h_{1i}\right)$ is contained in list $L_{1}$. If not, $C_{1}$ executes $H_{1}\;query$ first and then performs the following: (1) if the $ID_{i}\;\neq \;ID^{*}$ and $x_{i}\;\neq \;⊥$ (the public key has not been replaced), $C_{1}$ randomly selects $t_{i},h_{2i}\in Z_{q}^{*}$, calculates $T_{i}\;=\;t_{i}P,v_{i}\;=\;t_{i}\;+\;h_{2i}\left(x_{i}\;+\;d_{i}\right)$, and outputs signature $\tau_{i}\;=\;\left(T_{i},v_{i}\right)$; (2) if $ID_{i}\;=\;ID^{*}$ or $x_{i}\;=\;⊥$ (the public key is replaced), $C_{1}$ randomly selects $t_{i},v_{i}\in Z_{q}^{*}$, calculates $T_{i}\;=\;v_{i}P\;-\;h_{2i}\left(X_{i}\;+\;R_{i}\;+\;h_{1i}P_{pub}\right)$, and outputs signature $\tau_{i}\;=\;\left(T_{i},v_{i}\right)$. Then, $C_{1}$ updates $\left(m_{i},ID_{i},X_{i},R_{i},T_{i},h_{2i}\right)$ to list $L_{2}$.**Forgery**: Upon completing all queries, adversary $A_{I}$ outputs a forged signature $\tau^{*}\;=\;\left(T^{*},v^{*}\right)$ for $\left(ID^{*},m^{*}\right)$. If $ID_{i}\;\neq \;ID^{*}$, $C_{1}$ will abort and output ⊥. In line with the forking lemma, when $C_{1}$ replays $h_{1}$, $A_{I}$ produces another forged signature ${\tau^{*}}^{\left(2\right)}\;=\;\left(T^{*},{v^{*}}^{\left(2\right)}\right)$, resulting in two signatures sharing $T^{*}\;=\;t^{*}P$. According to the signature calculation, there are two equations:$$\left\{\begin{array}{c} v^{*(1)}\;\;=\;\;t^{*}\;+\;h_{2}^{*}\left(x^{*}\;+\;r^{*}\;+\;h_{1}^{*(1)}a\right)\\ v^{*(2)}\;\;=\;\;t^{*}\;+\;h_{2}^{*}\left(x^{*}\;+\;r^{*}\;+\;h_{1}^{*(2)}a\right) \end{array}\right.$$Considering the potential for public key replacement, $x^{*}$ remains unknown to $C_{1}$. The equation involves three unknown variables: $x^{*},t^{*},a$. The value of *a* can be determined using simultaneous equations as $a\;=\;\frac{v^{*(1)}\;-\;v^{*(2)}}{h_{1}^{*(1)}\;-\;h_{1}^{*(2)}}$. This approach enables $C_{1}$ to successfully employ $A_{I}$ as a subroutine to solve the ECDLP.Next, we calculate the probability that $C_{1}$ will win Game 1. First, we define three events: $\Delta_{1}$ indicates that $A_{I}$ does not exit during the query phase; $\Delta_{2}$ indicates that $A_{I}$ does not exit during the forgery; $\Delta_{3}$ indicates that $A_{I}$ successfully forges a valid signature. Therefore, the following are true:1. The probability that $A_{I}$ does not perform the ExtractPartialPrivateKey query for $ID^{*}$ during the query phase is $Pr\left[\Delta_{1}\right]\;=\;{\left(1\;-\;\frac{1}{q_{H_{1}}}\right)}^{q_{eppk}}$;2. The probability that $A_{I}$ does not exit in the forgery stage is the probability that $a_{i}$ outputs the signature corresponding to $ID^{*}$ in the forgery stage: $Pr\left[\Delta_{2}|\Delta_{1}\right]\;=\;\frac{1}{q_{H_{1}}}$;3. The probability of forging one valid signature is ε, and the probability of forging two valid signatures is $Pr[\Delta_{3}|\Delta_{2}\Delta_{1}]\;=\;\varepsilon$. Thus, $Pr\left[\Delta_{1}\Delta_{2}\Delta_{3}\right]\geq {\left(1\;-\;\frac{1}{q_{H_{1}}}\right)}^{q_{eppk}}\frac{1}{q_{H_{1}}}\varepsilon$, and challenger $C_{1}$ can solve an ECDLP hard problem with non-negligible probability. And because the difficult problem is actually unsolvable, the adversary $A_{I}$ cannot forge the signature. □

**Theorem 2.** 
*If the ECDLP hardness assumption cannot be solved in polynomial time, then our proposed enhanced scheme can effectively resist Type II attackers $A_{II}$. Specifically, suppose that the probability that a super Type II adversary $A_{II}$, being able to produce a valid forged signature in polynomial time, is ε. The maximum number of times that an adversary $A_{II}$ can query CreateUser query is $q_{H_{1}}$, the maximum number that an adversary $A_{II}$ can query ExtractSecretValue query is $q_{esv}$, and the maximum number that an adversary $A_{II}$ can query ExtractPartialPrivateKey query is $q_{rpk}$. The polynomial time algorithm $C_{2}$ has the advantage of $\varepsilon_{2}\;\geq \;{\left(1\;-\;\frac{1}{q_{H_{1}}}\right)}^{q_{esv}\;+\;q_{rpk}}\frac{1}{q_{H_{1}}}\varepsilon$ to solve difficult ECDLP problems.*


The proof steps of Theorems 2 and 1 are similar. However, it is necessary to mention that, in the proof of Theorem 2, ExtractPartialPrivateKey query and ReplacePublicKey query will not be initiated by $A_{II}$.

## 7. Performance Evaluation

In the previous section, we established the complete security and unforgeability of our proposed new scheme. We now turn our attention to the performance analysis of this new solution. To match the security level of the 1024-bit RSA algorithm, we employ the elliptic curve $y^{2}\;\;=\;\;x^{3}\;+\;\beta x\;+\;\gamma \;mod\;q,\;$$\beta ,\gamma \in Z_{q}^{*}$, with G being the order-*q* additive cyclic group on E(Fp). Here, *p* is a 512-bit prime, *q* is a 160-bit prime, and *P* serves as the generator of G. The implementation utilizes the open-source cryptography library MIRACL, running on a device equipped with an Intel(R) Core(TM) i5-3470 CPU @ 3.20 GHz x4 processor, 4GB of memory, and the 64-bit Ubuntu 22.04.3 LTS operating system. Table 1 presents the average running time of each basic cryptographic operation, measured over 1000 repeated experiments.

In the IoT environment, where device resources are limited and computing capacities are modest, there are stringent requirements for both computational efficiency and communication efficiency. In CLS schemes, computational efficiency is primarily determined by the computational load during the signing and verification phases, whereas communication efficiency largely depends on the size of the signature.

As illustrated in Table 2 and Figure 2, in terms of computational efficiency, during the signature phase, our scheme has a calculation time comparable to those of Feng, Ma et al.’s scheme, and is slightly lower than those of Karati, Pakniat, Du, Xiang, and other schemes. However, in the verification phase, our scheme has a calculation time of $3T_{sm}\;+\;3T_{pa}\;\;=\;\;1.008945\;\mathrm{ms}$, which is lower than Feng, Ma et al.’s calculation time of $4T_{sm}\;+\;3T_{pa}\;\;=\;\;1.343218\;\mathrm{ms}$, and Du, Xiang et al.’s calculation time of $4T_{sm}\;+\;2T_{pa}\;\;=\;\;1.341176\;\mathrm{ms}$, and is comparable to Karati et al.’s scheme. The total calculation time of our scheme is slightly lower than that of Karati et al.’s scheme, about 19.76% lower than Feng et al. and Ma et al.’s scheme, about 19.80% lower than Du et al. and Xiang et al.’s schemes, and about 20.05% lower than Pakniat et al.’s scheme, marking a significant improvement in computational efficiency.

Regarding communication efficiency, our scheme’s output signature size, $|G|\;+\;|Z_{q}^{*}|\left(360\;+\;120\;=\;480\;bits\right)$, is comparable to Feng, Ma, Karati, Du, and Xiang et al. Compared to Pakniat et al.’s scheme’s output signature size, $2|G|\;+\;|Z_{q}^{*}|\left(720\;+\;120\;=\;840\;bits\right)$, our scheme is more efficient in communication efficiency.

Our performance analysis shows that our scheme offers higher security and computational efficiency compared to Ma, Karati, Pakniat, Du, and Xiang et al.’s schemes. Compared with Feng et al.’s scheme, our scheme has improved computational efficiency while maintaining the original security level. Therefore, our scheme is a more suitable solution for the IoT environment.

## 8. Conclusions

The design of lightweight, secure, efficient, and reliable signature schemes for the IoT has become a hot research topic. Certificateless signature schemes have become an important solution for providing security services such as identity authentication, data integrity protection, and non-repudiation for the IoT due to their advantages of not requiring certificates and avoiding the key escrow problem. We review four PF-CLS schemes and use a novel attack method to attack the existing ones on four PF-CLS schemes with the same security vulnerabilities to prove that none of the four schemes can protect against Type I adversary forgery attacks. Subsequently, we propose an improved scheme, which is proven to be secure, reliable, and unforgeable under the ROM. Finally, the performance analysis of the new scheme shows that our scheme offers higher computational efficiency compared to the other PF-CLS schemes, making it more suitable for resource-constrained IoT environments.

Future research will focus on the following directions: (1) Existing schemes may expose user identities, and anonymous signatures can instead be realized by combining zero-knowledge proof (ZKP) or ring signature technologies. (2) Deploy the new scheme in practical resource-constrained IoT environment applications, and evaluate its feasibility during actual deployment, as well as its resistance to side-channel attacks such as timing attacks and power analysis.

## Figures and Tables

**Figure 1 sensors-25-05224-f001:**
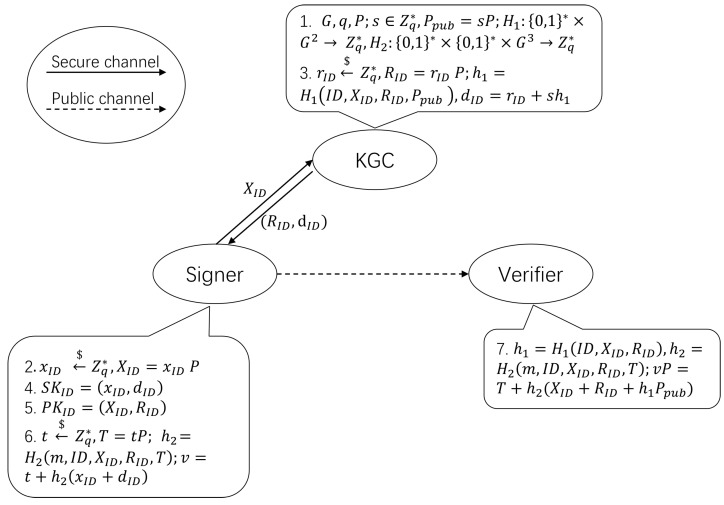
Our CLS scheme.

**Figure 2 sensors-25-05224-f002:**
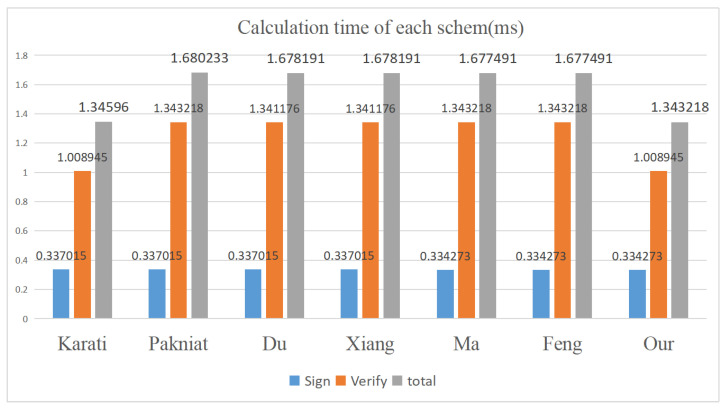
Calculation time of each scheme.

**Table 1 sensors-25-05224-t001:** Running time of cryptographic operations.

Notation	Operation	Time (ms)
$T_{sm}$	A scalar multiplication operation on elliptic curve	0.334273
$T_{pa}$	A point addition operation on elliptic curve	0.002042
$T_{m}$	A modular multiplication operation	0.000864
$T_{a}$	A modular addition operation	0.000455
$T_{inv}$	A modular inversion operation	0.002742
$T_{h}$	A general hash operation	0.002440

**Table 2 sensors-25-05224-t002:** Comparative analysis of performance.

Scheme	Sign	Verify	Signature Size	Against $A_{I}$	Against $A_{II}$
Karati [13]	$T_{sm}\;+\;T_{inv}$	$3T_{sm}\;+\;3T_{pa}$	$|G|\;+\;|Z_{q}^{*}|$	Insecure	Normal
Pakniat [14]	$T_{sm}\;+\;T_{inv}$	$4T_{sm}\;+\;3T_{pa}$	$2|G|\;+\;|Z_{q}^{*}|$	Insecure	Normal
Du [16]	$T_{sm}\;+\;T_{inv}$	$4T_{sm}\;+\;2T_{pa}$	$|G|\;+\;|Z_{q}^{*}|$	Insecure	Super
Xiang [19]	$T_{sm}\;+\;T_{inv}$	$4T_{sm}\;+\;2T_{pa}$	$|G|\;+\;|Z_{q}^{*}|$	Insecure	Super
Ma [20]	$T_{sm}$	$4T_{sm}\;+\;3T_{pa}$	$|G|\;+\;|Z_{q}^{*}|$	Super	Insecure
Feng [21]	$T_{sm}$	$4T_{sm}\;+\;3T_{pa}$	$|G|\;+\;|Z_{q}^{*}|$	Super	Super
Our	$T_{sm}$	$3T_{sm}\;+\;3T_{pa}$	$|G|\;+\;|Z_{q}^{*}|$	Super	Super

## Data Availability

The original contributions presented in this study are included in the article. Further inquiries can be directed to the corresponding author(s).

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
