# Peer review of "A New Efficient and Provably Secure Certificateless Signature Scheme Without Bilinear Pairings for the Internet of Things"

_sensors, 2025, doi:10.3390/s25175224_

Round 1
Reviewer 1 Report
Comments and Suggestions for Authors
In this paper, the authors present a new pairing-free certificateless scheme for IoT applications. The authors claim that their proposal is secure against type I and type II adversary attacks and compare their proposal with other schemes available in previous work.
The structure of the paper is appropriate. There are abundant manuscripts about the topic with a similar structure. Nevertheless, the manuscript has several problems with the usage of English. There are typos and some grammar problems.
The authors analyze Xiang et al, Du et al, Karati et al, Pakniat and Vanda and Ma et al. They provide a summary of these schemes and carry out an analysis of their vulnerabilities. Previous work has already shown that Xiang et al and Du et al are vulnerable to type I adversary attacks. This is already stated in the manuscript. Similarly, Shim has already proved the same for Karati et al and Pakniat and Vanda. Although the authors propose another approach to prove these vulnerabilities, they get to an already known point. Regarding Ma et al, previous work has already proved that this scheme is vulnerable to type II adversary attacks (see Feng et al “Blockchain-enhanced efficient and anonymous certificateless signature scheme and its application”, In Pervasive and Mobile Computing, Vol. 105, 2024) So the contributions here are not relevant. Thus:
- Section 1.2 needs to be reformulated.
 - Previous work needs to be revised because there are advances that have not been taken into account.
 - Section 3 details what is already available in previous work, so it is not actually necessary.
 - In general, I would suggest to limit the contents of the paper to the contributions of the authors.

Section 2 needs some improvements also. Please describe games #1 and #2 properly before the initialization phases. Also, there are some typos with adversary ids in game #2.
I cannot fully assess the proposed CLS scheme because I am not a cryptography expert. Nevertheless, it is necessary to provide details about the secure channel between the signer and the KGC, given the fact that this is a critical component and that the system has been designed for IoT applications. Also, it is necessary to explain how that proposal fits in the security model described in section 2.2, as this is key to fully understand proof in section 6. Query phases in section 6 are difficult to follow. I recommend rewriting them to improve clarity.
Regarding the performance evaluation:
- Table 1 and Figure 2 contain the same information. I recommend to erase Figure 2.
 - Given the fact that the proposal is for IoT applications, it would have been advisable to perform the experiments with an IoT device. The experiments seem to have been carried out on a regular computer.
 - Comparing schemes with running times is this type of papers is very frequent. But it is important to consider that running times depend on the characteristics of the system and the quality of the software. Thus, I would try to avoid the term “computational cost” which, in my opinion, is not accurate in this case.
 - Table 2 seems inconsistent with previous work (e.g. signature size of Pakniat et al.)

I would also suggest to mention some future works in section 8.
Comments on the Quality of English LanguageThe manuscript has several problems with the usage of English. There are typos and some grammar problems.
Reviewer 2 Report
Comments and Suggestions for Authors
In this paper, authors systematically reviews, attack, and improve pairing-free certificateless signature schemes tailored for resource-constrained IoT environments. The authors expose vulnerabilities in five state-of-the-art PF-CLS schemes, particularly under Type I and Type II adversarial models. Building on this analysis, they design a new scheme resistant to both adversarial types and support its robustness with security proofs under the random oracle model. Performance is evaluated in terms of computational and communication efficiency using real cryptographic benchmarks.
The paper is rich in technical content and provides a meticulous analysis of both legacy schemes and the proposed improvement. The adversarial models are clearly defined and well-applied to the selected schemes. However, the paper suffers from several stylistic inconsistencies and structural redundancies. At times, it relies heavily on formulaic steps that make readability cumbersome, especially in the cryptanalysis section. Presentation structure should be enhanced. There are many long sentences and they need to be divided.
Some comments:
- Which paper are you referring to in abstract? "for a CLS scheme designed by Ma et al." It is better to avoid giving references in abstract but rather concentrate on main achievements
- Give reference in "We analyze Ma et al.’s scheme " on page 3 line 92
- Why do you call "In this chapter," in the beginning of Section 3. It is not book chapter but section
- Present fig. 2 in a better format.
- This sentence in conclusion section is too long and not understandable, "Moreover, for the certificateless signature scheme recently proposed by Ma et al. for the  IoT, although Ma et al.’s scheme improved the Du et al. and Xiang et al.’s schemes by  using a hash function to tightly link the two parts of the public key during public key  generation, thus defending against Type I adversary forgery attacks, Ma et al.’s scheme  overlooks forgery attacks by Type II adversary."
- Although theoretical security is shown via reduction to ECDLP, practical security aspects (e.g., side-channel resistance) and deployability are not discussed.
-  The use of the Forking Lemma is appropriate but lacks citations or a brief reminder of its core idea. A simplified flow diagram of the reduction would help general readers.
-  No statistical variability (e.g., standard deviation across runs) is provided. Use of a single system and library may reduce generalizability.
- Many sections rely on repetitive phrases like “we prove that” or “we use the same attack method.” This makes the prose feel mechanical.
- Section 1.1: Reference list is long but lacks critical analysis—consider a comparative timeline figure.
- Section 3: Repeated phrase “refer to Xiang et al.’s scheme” could be replaced with tabular summaries for clarity.
- Figure 2 and 3 are not captioned appropriately. Their value is unclear.
- The term “super adversary” is confusing and should be replaced with standard “adaptive adversary.”
- Security Theorem Proofs are verbose, repetitive, and should be modularized with diagrams.
- The abstract and introduction could better highlight why now is the right time for a new PF-CLS scheme—e.g., improvements in ECC libraries or attack sophistication.
- The introduction references prior schemes (e.g., Ma et al., Xiang et al.) but doesn't clearly indicate what makes them foundational or widely adopted. Add stronger commentary on their practical usage or industry relevance.
- Typos: “Performancec” in Section 7 title should be corrected.
Reviewer 3 Report
Comments and Suggestions for Authors
1) The introduction would benefit from a clearer, more focused statement of how the proposed scheme advances beyond Ma et al. (2023) and other recent PF‑CLS constructions. Please explicitly summarize the unique features and advantages of your approach.
2) While the security proofs under the Random Oracle Model are detailed, it would strengthen the exposition to more explicitly delineate which queries (e.g., ExtractSecretValue vs. ReplacePublicKey) map to Type I and Type II adversary capabilities.
3) Sections 4.1–4.3 propose novel attack methods against prior schemes. To aid comprehension, consider adding a short illustrative example or flowchart for one representative attack, highlighting key steps and assumptions.
4) The comparison in Table 2 demonstrates efficiency gains, but it is unclear whether all schemes were implemented under identical cryptographic libraries and hardware. Please clarify your experimental setup and, if possible, report variance (e.g., standard deviation) over multiple runs to demonstrate measurement stability.
5) Some algorithmic descriptions (e.g., in Sections 3.2 and 3.4) closely mirror one another, which may confuse readers. I recommend consolidating common steps into a single overview and highlighting only the differences for each scheme, thereby reducing repetition.
Reviewer 4 Report
Comments and Suggestions for Authors
This paper studied a new efficient and provably secure certificateless signature scheme without bilinear pairings for IoT. It looks interesting and sure this paper will be a better reference for researchers who have been studying this. However, I decided to make a major revision to this paper for the following reasons,
- The authors mention the Random Oracle Model (ROM) in several proofs, but the assumptions under which ROM holds and how it applies to the real-world deployment of the proposed CLS scheme should be more explicitly discussed in Section 6.

- While the paper provides detailed cryptanalysis of five PF-CLS schemes, including novel attack methods, it would strengthen the work if the authors could briefly assess the real-world feasibility or cost of executing such attacks, especially in constrained IoT environments.

- Section 6 provides rigorous security proofs, but readers may benefit from a comparative table summarizing the complexity assumptions and reduction tightness between the proposed scheme and prior works (e.g., Karati, Ma, Du).

- The term "Performancec evaluation" in Section 7 appears to be a typographical error. Please revise to “Performance Evaluation.” A thorough proofreading for minor grammatical or formatting inconsistencies across section titles and equations is recommended.

- Although the scheme introduces H1 and H2 as secure hash functions, it is unclear whether these are modeled as independent or jointly secure functions. Clarifying their construction or potential instantiations would improve practical understanding and reproducibility.

- The conclusion effectively summarizes the paper’s contributions, but it could be enhanced by briefly reflecting on limitations and suggesting specific future research directions, such as quantum resistance or integration with authentication protocols.

- In Table 1 and Figure 2, execution times for cryptographic primitives are presented without statistical measures (e.g., standard deviation or confidence intervals). Adding these would improve the robustness and reliability of the performance evaluation.

Round 2
Reviewer 2 Report
Comments and Suggestions for Authors
Authors have responded to my previous comments. Some minor comments.
- Remove caption in fig 2, there is already a typo there as well
- Section 6 starts with lemma. Make introductory paragraphs for each section
- You still say chapter in the beginning of Section 5
Reviewer 3 Report
Comments and Suggestions for Authors
All my previous comments have been addressed.
Reviewer 4 Report
Comments and Suggestions for Authors
I have carefully reviewed the revised version of your manuscript “A New Efficient and Provably Secure Certificateless Signature Scheme without Bilinear Pairings for IoT.”
In my earlier review, I requested substantial clarifications and improvements, particularly regarding the role of the Random Oracle Model in your proofs, the practical feasibility of proposed attacks in IoT settings, a comparative summary of security assumptions, clarification of the independence of hash functions, and more careful proofreading. I also suggested that the conclusion should highlight limitations and directions for future research, and that the performance evaluation should include more robust statistical reporting.
I am pleased to see that all of these points have been properly addressed in your revision. The current version provides clearer explanations of the ROM-based proofs, a stronger discussion of the feasibility of attacks, and an effective comparative table that situates your work relative to prior schemes. The presentation has been significantly polished, terminology is consistent, and the treatment of the two hash functions is now more explicit. The conclusion has been strengthened with thoughtful reflections on limitations and future directions such as quantum resistance and anonymous signatures. Finally, the performance evaluation now includes averaged results and comparative analysis that convincingly demonstrate the efficiency of your scheme.
Overall, the revised manuscript is technically solid, well-written, and makes a clear contribution to the field of certificateless signature schemes for IoT. I therefore recommend acceptance of this paper for publication.